# Training-Free Retrieval-Augmented Generation for Knowledge-Intensive Visual Question Answering

## Abstract

Recent advancements in multimodal large language models (MLLMs) have achieved strong performance in vision-language tasks such as visual question answering (VQA). However, these models struggle with knowledge-intensive VQA (KI-VQA) tasks that require fine-grained domain knowledge, as seen in benchmarks such as Encyclopedic VQA and InfoSeek. To address these challenges, we propose a novel retrieval-augmented generation (RAG) framework, referred to as KIRA, designed to enhance the capability of MLLMs for KI-VQA without task-specific fine-tuning. Our target is to integrate general image-text similarity with detailed knowledge context to achieve precise entity recognition. To this end, we leverage CLIP to obtain general image-text matching, and design a verification mechanism according to detailed question-text relevance to improve recognition accuracy. We evaluate our method on KI-VQA benchmarks, demonstrating significant improvements of 47.5% on Encyclopedic VQA and 16.2% on InfoSeek, all achieved without additional training. These results highlight the potential of our training-free, plug-and-play framework for solving knowledge-intensive visual question answering tasks.

## 1 Introduction

Recent advancements in multimodal large language models (MLLMs) (OpenAI, 2023; Li et al., 2022; Dai et al., 2023; Liu et al., 2023b;a; Lin et al., 2023b; Gao et al., 2024) have shown promising performance in various vision-language tasks, including visual question answering (VQA), visual grounding, and image captioning. Despite the achievements, current MLLMs typically focus on answering questions requiring limited outside knowledge (e.g., commonsense knowledge), and hence struggle with knowledge-intensive VQA tasks such as Encyclopedic VQA (Mensink et al., 2023) and infoseek (Chen et al., 2023).

Knowledge-intensive visual question-answering (KI-VQA) is distinct from VQA tasks relying on commonsense knowledge such as OK-VQA (Marino et al., 2019) in that the knowledge required for answering questions is in a very fine-grain level. This adds significant complexity to the task, as identifying the relevant information often demands precise recognition of specific entities within the image. As illustrated in Figure 2, the model is required to recognize the "Amazon Arena" in the image and process the knowledge about its sustainability feature. As shown by studies (Chen et al., 2023; Mensink et al., 2023; Vrandečić & Krötzsch, 2014), existing state-of-art MLLMs still struggle with providing accurate answers in such specialized contexts due to the lack of specialized knowledge in those models, limiting their applicability in real-world scenarios.

For the knowledge-intensive VQA tasks, a promising strategy is to utilize an external knowledge base, which not only avoids the high cost of encoding knowledge into model parameters via fine-tuning but also provides more interpretability by separating the retrieval and answer generation processes. Retrieval-augmented generation (RAG) has been proposed as a key technique to retrieve relevant knowledge from an external source to support the answer generation process. However, despite the encouraging performance of RAG in unimodal tasks such as those in natural language processing (NLP) (Rubin et al., 2021; Xiong et al., 2020), its application to multimodal tasks remains challenging.

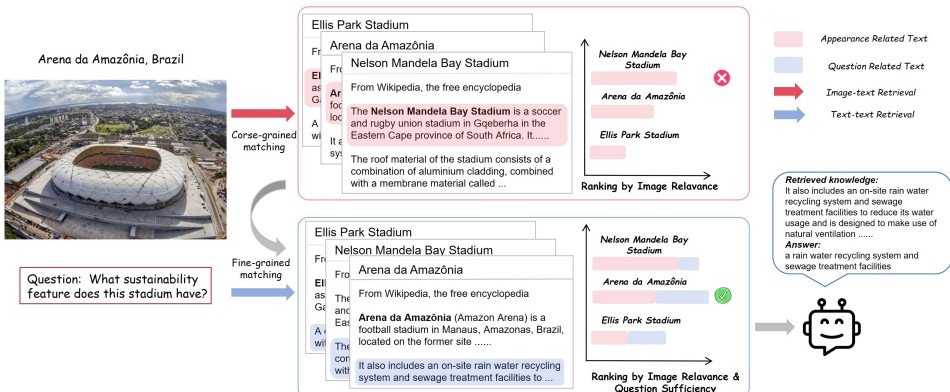

Figure 1: **We show a knowledge-intensive VQA sample and the core idea of this paper.** To correctly answer the question, detailed specialized outside knowledge is required. We retrieve such knowledge in a two-step manner. Coarse-grained matching finds relevant knowledge via image-text retrieval models and the results are improved in fine-grained matching where text retrieval models are used to evaluate the sufficiency of retrieved knowledge.

In this work, we focus on the multimodal RAG framework targeting Knowledge-Intensive Visual Question Answering (KI-VQA). Two main strategies have been explored to tackle multimodal knowledge retrieval in previous work. The first strategy involves end-to-end training on large multimodal datasets, enabling models to learn both entity recognition and retrieval of relevant text from the knowledge base. However, the training requires extensive resources such as more than 1,000 TPU hours for REVEAL (Hu et al., 2022) and suffers from poor generalization due to task-specific fine-tuning. The second strategy accomplishes the task using two-stage strategy. The methods leverage the multimodal retrieval model CLIP Radford et al. (2021) for entity recognition avoiding the need for costly fine-tuning, then extracting relevant text according to recognition results. Systems such as WikiLLaVA (Caffagni et al., 2024) adopt this approach. However, this method relies on global image-text matching for recognition, which can miss crucial fine-grained details and result in imprecise recognition results.

To address these limitations, we propose a novel multimodal RAG framework for knowledge-intensive visual question-answering, referred to as Knowledge-intensive Retrieval Augmentation(KIRA), to achieve precise knowledge retrieval. As illustrated in Figure 2, we integrate both general image-text similarity and details in knowledge context to achieve precise entity recognition, while avoiding fine-tuning. We leverage CLIP to provide initial recognition according to general image-text matching, then we design a verification mechanism according to detailed question-text relevance. The verification mechanism is designed based on the following hypothesis: if the associated knowledge context of the recognition result is not sufficient to answer the question, it is likely either the recognition result is incorrect, or the current knowledge base is unable to provide enough information. Using this verification mechanism, we inject knowledge details into the recognition process.

Specifically, our framework consists of three core components: an entity recognition module, a relevant context extraction module, and an answer generation module. First, the entity recognition module combines general text-image similarity with knowledge context details to accomplish fine-grained entity identification. Subsequently, the relevant context extraction module retrieves knowledge based on the recognized entities. After that, we complement this with additional information that is not directly related to entities within the image. Finally, the answer generation module employs an MLLM to generate an answer from the retrieved knowledge contexts.

We demonstrate the effectiveness of proposed methods on knowledge-intensive VQA benchmark Encyclopedic VQA (Mensink et al., 2023) and infoseek Chen et al. (2023). We achieve significant improvements on both datasets compared with baseline models without any training, such as 47.5% on EVQA and 16.2% on the InfoSeek.

Our main contributions are summarized as follows:

- We are the first to propose a plug-and-play training-free retrieval-augmented generation framework to solve knowledge-intensive visual question-answering tasks. We consider our methods as a good starting point for further exploration.

- Our design achieves precise entity recognition by integrating general text-image similarity with knowledge context details, which guarantees the retrieval performance in a fine-grained knowledge base.

- The experimental results on two popular knowledge-intensive benchmarks demonstrate the superiority of the proposed methods. We achieve impressive improvements without any training.

## 2 RELATED WORKS

### 2.1 INFORMATION RETRIEVAL MODELS

The landscape of information retrieval models encompasses a wide range of approaches. In the domain of text retrieval, works (Rubin et al., 2021; Xiong et al., 2020; Karpukhin et al., 2020) have been developed to facilitate retrieval for open-domain question answering. ColBERT (Khattab & Zaharia, 2020; Santhanam et al., 2021) employs a late interaction approach, where BERT (Devlin et al., 2018) embeddings are computed for both queries and documents, with interaction performed at a later stage. Contrary to traditional retrievers that use early interaction, Contriever (Izacard et al., 2021) incorporates interaction between query and document representations at a later stage, achieving competitive performance in large-scale retrieval scenarios.

Other efforts (Frome et al., 2013; Faghri et al., 2017) have been made in the cross-modal retrieval domain, particularly in image-text retrieval, which has seen significant advancements over the years. DeViSE (Frome et al., 2013) was a pioneering approach that projected images and words into a shared embedding space using deep neural networks, leveraging pre-trained word vectors to capture semantic relationships. The introduction of CLIP (Radford et al., 2021) marked a significant leap forward in the field by training on a large-scale dataset of images and their corresponding textual descriptions from the internet. More recently, works (Lin et al., 2023a) propose to train a multi-modal Retrieval through fine-grained late-interaction alignment. In this paper, we propose a training-free multi-modal retrieval framework for visual question answering by incorporating both types of retrieval models mentioned above. This integrated approach aims to enhance the performance and versatility of retrieval systems in complex, multi-modal tasks.

### 2.2 MULTI-MODAL RETRIEVAL-AUGMENTED GENERATION

Recently, retrieval-augmented generation (RAG) (Lewis et al., 2020; Nakano et al., 2021; Borgeaud et al., 2021; Yu et al., 2021) has been proposed to enhance large language models (LLMs) by incorporating knowledge from external databases, thereby improving performance on knowledge-intensive tasks and reducing hallucination. REALM (Lewis et al., 2020) expands the input space with relevant text passages retrieved from external sources, while WebGPT (Nakano et al., 2021) enables models to search and navigate the web for additional information. In the context of vision-language tasks, previous works such as REVEAL (Hu et al., 2022) and KAT (Gui et al., 2021) have explored retrieval-augmented approaches using vision-language models (VLMs) for knowledge-intensive visual question answering (VQA) by training a generative vision-language model and a multi-modal retriever. More recently, Wiki-LLaVA (Caffagni et al., 2024) has explored RAG in popular multi-modal LLMs with the Wikipedia knowledge base, targeting challenging knowledge-intensive benchmarks through fine-tuning. However, the results reveal that the main challenge is the accuracy of knowledge retrieval instead of the MLLM's ability to read retrieved articles. EchoSight (Yan & Xie, 2024) trains a reranking module to improve the visual-only retrieval and achieve promising improvements in KVQA. Compared with EchoSight, we adopt a more challenging setting that does not include any fine-tuning and considers a text-only knowledge base. In this paper, we focus on exploring a training-free RAG approach to tackle knowledge-intensive VQA tasks, significantly boosting the performance of multi-modal large language models (MLLMs).

## 2.3 Knowledge Visual Question Answering

Knowledge Visual Question Answering (KVQA) has emerged as a crucial task for evaluating the ability of vision-language models to integrate external knowledge sources. The OKVQA benchmark (Marino et al., 2019) stands as one of the pioneering initiatives to explicitly necessitate the use of external knowledge. It comprises questions that cannot be answered solely based on image content, demanding information from external sources like Wikipedia or general world knowledge. Building on the foundation laid by OKVQA, AOKVQA (Schwenk et al., 2022) was introduced to further augment the scope and complexity of VQA tasks involving external knowledge. Another notable benchmark, Knowledge-enriched VQA (KVQA) (Shah et al., 2019), delves into questions requiring knowledge about named entities such as people, places, organizations, and events. Recent endeavors such as Encyclopedic VQA (Mensink et al., 2023) and InfoSeek (Chen et al., 2023) have pushed the boundaries of standard knowledge-based VQA by posing queries that demand in-depth knowledge about specific entities. Even large language model based models struggle to perform adequately on these tasks without retrieving information from external sources. In this paper, we focus primarily on the most challenging benchmark, Encyclopedic VQA, to evaluate the effectiveness of our proposed framework.

## 3 Methods

In this section, we introduce the proposed KIRA framework. The proposed method is designed to perform training-free retrieval from a fine-grained specialized knowledge base, and then accomplish a knowledge-intensive VQA task. In Section 3.1, we systematically formulate the problem we aim to solve. Section 3.2 outlines an Entity Recognition module to identify the entity within the image. Section 3.3 details the Relevant Context Extraction module, where the useful knowledge context is retrieved based on the entity recognition results. Finally, Section 3.4 describes the Answer Generation module.

### 3.1 Task Formulation

Formally, the visual question answering dataset is defined as $\mathbb{D} = \{(v_i, q_i, a_i) \mid i = 1, 2, \ldots, N\}$ where $v_i$ to denote the $i^{th}$ image, $q_i$ and $a_i$ to denote the $i^{th}$ question and its corresponding answer respectively. An external knowledge base is denoted as $\mathbb{K} = \{E_j \mid j = 1, 2, \ldots, M\}$, where $N$ is the number of entity articles contained in the knowledge base. For the $j^{th}$ entity article, we dived the article into $M_j$ text snippets, therefore $E_j = \{t_1^j, \ldots, t_{M_j}^j\}$, where $t_m^j$ denotes the $m^{th}$ text snippet.

### 3.2 Entity Recognition

We first present our Entity Recognition module. In this stage, we perform entity recognition considering both the general image-text similarity and the details in knowledge contexts. We employ a two-stage procedure. In the first stage, we construct a candidate set of entities by coarse-grained searching. After that, we apply fine-grained recognition to yield the final recognition results.

#### 3.2.1 Coarse-grained Searching

We first collect a small set of possible candidates according to the similarity between the image $v_i$ and a brief description of the predefined categories in the knowledge base, since it is too costly and inefficient to perform fine-grained matching on each entity in the knowledge base. Specifically, for a visual question answering task $\{(v_i, q_i) \mid i = 1, 2, \ldots, N\}$, we construct an entity candidate set leveraging CLIP. Specifically, we transfer images and brief descriptions for each entity in the knowledge base into vectors, then utilize the cosine similarity metric. We use the first text snippet in the article as a brief introduction to an entity.

$$I_i = \text{CLIPVisual}(v_i), \quad T_j^c = \text{CLIPText}(t_1^j) \tag{1}$$

$$\text{CoarseSim}(i, j) = \frac{I_i \cdot T_j^c}{\|I_i\|\|T_j^c\|} \tag{2}$$

For each image $v_i$, we collect the top $K_c$ best-matched entities from the knowledge base, i.e., $S_i = \{E_1, \ldots E_{K_c}\}$, where $|S_i| = K_c$, $K_c \ll M$.

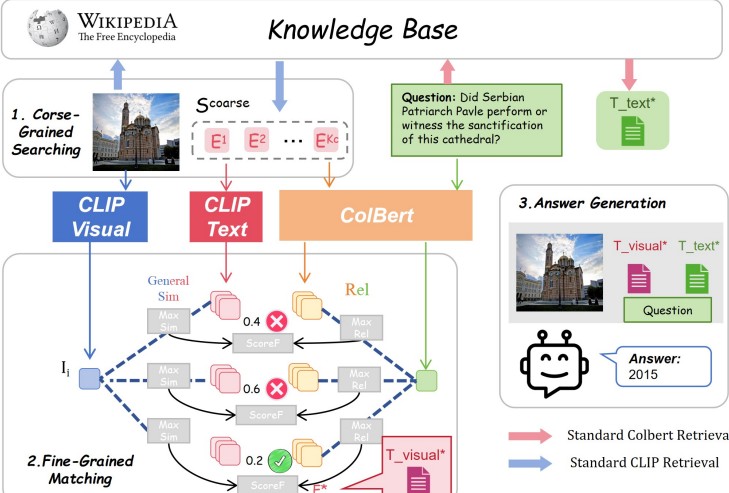

Figure 2: **The overall pipeline of the proposed framework.** Given a visual question that requires outside knowledge, we perform *Coarse-grained Searching* to extract an article subset from the WiKi knowledge base. To choose the informative article, *Fine-grained matching* evaluates the candidates from two aspects, image relevance, and question relevance. Finally, two text chunks are retrieved and fed to the MLLMs answer generation.

### 3.2.2 FINE-GRAINED MATCHING

**General Image-text Matching** In this procedure, we aim to measure the similarity between the image $v_i$ and knowledge articles in a fine-grained manner for collected candidates in $S_i$. For an article $E_j \in S_i$, we measure the similarity between the image $v_i$ and each text snippet $t_m^j \in E_j$, and use the closest similarity to denote the general image-text similarity between the image and the entity.

$$T_{(j,m)}^f = \text{CLIPText}(t_m^j) \tag{3}$$

$$\text{GeneralSim}(i, j) = \max_{m=1}^{M_j} \frac{I_i \cdot T_{(j,m)}^f}{\|I_i\|\|T_{(j,m)}^f\|} \tag{4}$$

**Detail Verification** For the KI-VQA task, it is necessary to consider the detailed information in the knowledge text to achieve accurate entity recognition. However, it is extremely difficult to directly compare the image with the detailed description in an article since knowledge texts are often concise and categorical, and differ significantly from the instance-specific descriptions typically found in image captions.

To inject detailed information in knowledge articles into the recognition procedure, we introduce a detailed characteristics check mechanism based on question-text relevance. Specifically, we hypothesize that if there is no sufficient knowledge context in the associated knowledge context to answer the question, it is either that the retrieved entity is incorrect, or the current knowledge base is insufficient to answer the question. Thus, we measure the relevance between knowledge article $E_j$ and the visual question $(v_i, q_i)$, ensuring that fine-grained details play a role during the retrieval process. Specifically, we measure the relevance of knowledge article $E_j$ and question $q_i$ using Colbert. Colbert takes a query and a set of test snippets as input and then measures the relevance of the query and each snippet. The process is as follows:

$$\text{Rel}(i, j) = \max_{t_m^j \in E_j} \text{ColBERT}(q_i, t_m^j) \tag{5}$$

Given $\text{GeneralSim}(i, j)$ and $\text{Rel}(i, j)$, we are able to find the best entity in $S_i^{coarse}$ with closest proximity with $(v_i, q_i)$. We obtain the final entity recognition results as follows:

$$\text{ScoreF}(i, j) = \frac{\lambda}{\text{GeneralSim}(i, j)} + \frac{1}{\text{Rel}(i, j)} \tag{6}$$

$$E_i^* = \underset{E_j \in S_i}{\arg\min}\, \text{ScoreF}(i, j) \tag{7}$$

where $\lambda$ is a hyperparameter that controls the trade-off between the GeneralSim and Rel.

### 3.3 RELEVANT CONTEXT EXTRACTION

In this stage, we perform relevant context extraction provided the entity recognition results. Our relevant context extraction procedure consists of two parts: the visual-related context and the text-related context. For the visual-related context, we retrieve useful texts from the associated knowledge context of the previously retrieved entity. The specifical process is as follows using Colbert:

$$\text{T\_visual}_i^* = \underset{t_v \in E_i^*}{\arg\max}\, \text{ColBERT}(q_i, t_v) \tag{8}$$

After that, we retrieve the text-related context is considered as compensation for visual-related context. The text-related context is retrieved from the whole knowledge base only according to the question. Such text-related context plays a role in the circumstance where the question requires not only knowledge about the entity appearing within the image but also information about entities outside the image.

$$\text{T\_text}_i^* = \underset{t_l \in E_1 \cup \ldots \cup E_M}{\arg\max}\, \text{ColBERT}(q_i, t_l) \tag{9}$$

### 3.4 ANSWER GENERATION

Given visual question $(v_i, q_i)$ and previously obtained knowledge texts $\text{T\_visual}_i^*$ and $\text{T\_text}_i^*$, we utilize an off-the-shelf Multimodal Large Language Model(MLLM) to generate the final answer.

$$\hat{a}_i = \text{MLLM}(v_i, q_i, [\text{T\_visual}_i^*, \text{T\_text}_i^*]) \tag{10}$$

The MLLM is equipped with essential knowledge context for knowledge-intensive question answering, enabling the system to handle complex questions that demand precise and specialized knowledge. The detailed prompt template is shown in the Appendix visualization.

## 4 EXPERIMENTS

In this section, we introduce the experimental results on challenging benchmarks and provide implementation details. Moreover, we provide comprehensive ablation studies to demonstrate the effectiveness of our method.

### 4.1 EVALUATION BENCHMARKS

**Encyclopedic VQA.** To evaluate the performance of multi-modal large language models (MLLMs) on visual questions requiring extensive external knowledge, we utilize the recently proposed Encyclopedic VQA Mensink et al. (2023) dataset. This dataset contains visual questions about detailed properties of fine-grained categories and is primarily constructed using annotations from iNaturalist 2021 Horn et al. (2021) and the Google Landmarks Dataset V2 Weyand et al. (2020). The Encyclopedic VQA dataset comprises approximately 221k question-answer pairs associated with 16.7k different fine-grained entities, each represented by up to five images. The dataset is divided into training, validation, and test splits, containing 1M, 13.6k, and 5.4k samples, respectively. For the knowledge base, Encyclopedic VQA filters out non-English Wikipedia pages from the WIT dataset Srinivasan et al. (2021) and compiles a total of 2M Wikipedia pages. Since our framework focuses on a training-free setting, we utilize a knowledge base consisting of relevant Wikipedia pages associated with iNaturalist and Google Landmarks. Specifically, our knowledge base includes the Wikipedia pages from the train, test, and validation sets of the Encyclopedic VQA dataset, comprising a total of 18,000 unique articles. We report the BEM (Balanced Evaluation Metric) Bulian et al. (2022) score of the test set using official scripts.

Table 1: **The performance comparison on Encyclopedic VQA and the InfoSeek benchmark.** The "*KB*" indicates the knowledge base type. "-" means no knowledge base. "*T*" is a text-only knowledge base. "*T&V*" means a knowledge base with both image and text. For InfoSeek, the *Unseen-Q and Unseen-E* means the unseen questions and entities category.

| Method | LLM | KB | Retrieval | EVQA | | InfoSeek | | |
| | | | | Single-hop | All | Unseen-Q | Unseen-E | All |
|---|---|---|---|---|---|---|---|---|
| **Fine-tuned** | | | | | | | | |
| LLaVA-1.5 | Vicuna-7B | - | - | 23.3 | 28.5 | 19.4 | 16.7 | 17.9 |
| Wiki-LLaVA | Vicuna-7B | T | KB Sentences | 21.8 | 26.4 | 26.6 | 24.6 | 25.5 |
| DPR | Multi-passage BERT | T&V | KB Sentences | 29.1 | - | - | - | 12.4 |
| EchoSight | LLaMA3-8B | T&V | KB Section | 38.9 | - | - | - | 31.3 |
| **Training-free** | | | | | | | | |
| Vanilla | PaLM | - | - | 19.7 | - | 5.1 | 3.7 | 4.3 |
| | LLaMA3-8B | - | - | 13.4 | 10.6 | 1.7 | 0.9 | 1.2 |
| BLIP-2 | Flan-T5$_{XL}$ | - | - | 12.6 | 12.4 | 12.7 | 12.3 | 12.5 |
| InstructBLIP | Flan-T5$_{XL}$ | - | - | 11.9 | 12.0 | 8.9 | 7.4 | 8.1 |
| LLaVA-1.5 | Vicuna-7B | - | - | 16.3 | 16.9 | 9.6 | 9.4 | 9.5 |
| BunnyV1.1 | LLaMA3-8B | - | - | 36.3 | 25.4 | 12.8 | 12.3 | 12.5 |
| Google Lens | PaLM | T&V | KB Section | - | **48.8** | - | - | - |
| | GPT-3 | T&V | KB Section | - | 44.6 | - | - | - |
| **FRA** | | | | | | | | |
| Vanilla | LLaMA3-8B | T | KB Sentences | **51.0** | 47.5 | 16.7 | 14.0 | 15.2 |
| Bunny-1.1 | LLaMA3-8B | T | KB Sentences | 48.4 | 47.0 | **18.2** | **14.7** | **16.2** |

**InfoSeek.**  The InfoSeek (Chen et al., 2023) benchmark is tailored for information-seeking questions that require expert knowledge. It consists of 1.3 million visual information-seeking questions, encompassing more than 11,000 visual entities from OVEN (Hu et al., 2023). The questions in this dataset are diverse, and the answers can be referenced from Wikipedia. For the knowledge base, Infoseek offers a knowledge base with 100,000 Wikipedia articles accompanied by images. Under our training-free setting, our knowledge base includes the Wikipedia pages from the train, and validation sets of the InfoSeek dataset, comprising a total of 6,576 unique articles. Since ground truth for the test split is not available, we report the VQA score on the validation split with official scripts.

## 4.2 Implementation Details

For pre-trained retrieval models, we employ the CLIP Radford et al. (2021) ViTL/14@336 variant following previous works Caffagni et al. (2024). The dense text retrieval model is set to ColBERTv2 Santhanam et al. (2021). For hyper-parameters, we use the validation set of Encyclopedic VQA to select hyper-parameter selection. For the Infoseek, we randomly sample 1,000 data from the validation set since only the validation set is available. The $\lambda$ is set to 64 for Encyclopedic VQA and 256 for InfoSeek benchmark. The number of entities selected during coarse-grained matching $K_c$ is 20.

## 4.3 Model Evaluation

**Baselines.**  To demonstrate the effectiveness of our proposed methods, we compare KIRA with two kinds of baselines. The first category is *Training-free methods*, which means the model is not fine-tuned on the training set of E-VQA or InfoSeek. We report the performance of BLIP-2 Li et al. (2023), InstructBLIP Dai et al. (2023), Bunny-1.1 He et al. (2024), LLaVA1.5 Liu et al. (2023a) to show the knowledge encoded in the models' parameters. The vanilla means the language model generates the answer based on the question only. As suggested in Mensink et al. (2023), we include pre-trained PaLM (Chowdhery et al., 2022) and GPT-3 (Brown et al., 2020) with Google Lens Google (2023) as baselines, where Google Lens is a powerful retrieval tool that identifies image content by comparing query images with those in its database and the best matching knowledge base article for predicted entity is used as the retrieval result. The second category is *Fine-tuned* methods, which utilize the training set of E-VQA or InfoSeek to improve the retrieval or answer generation ability. DPR (Lerner et al., 2024) is an Entity Retrieval system trained for visual question answering.

Table 2: **Framework design ablation studies on the Encyclopedic VQA and the InfoSeek**. We report the top K article-level recall and the detailed VQA category performance to showcase the effectiveness of each design.

| Method | Recall@K | | | | VQA Performance | | | |
|---|---|---|---|---|---|---|---|---|
| | K=1 | K=5 | K=10 | K=20 | Single-hop | Multi-hop | Unseen-Q | Unseen-E |
| **Encyclopedic VQA** | | | | | | | | |
| CLIP I-T | 21.0 | 44.2 | 57.9 | 70.1 | - | - | - | - |
| + FGM | 43.4 | 60.2 | 65.1 | 70.1 | 49.9 | 33.2 | - | - |
| + TRC | - | - | - | - | 51.0 | 41.1 | - | - |
| **InfoSeek** | | | | | | | | |
| CLIP I-T | 30.2 | 47.8 | 51.5 | 56.3 | - | - | - | - |
| + FGM | 32.1 | 46.4 | 46.8 | 56.3 | - | - | 17.3 | 13.9 |
| + TRC | - | - | - | - | - | - | 18.2 | 14.7 |

Wiki-LLaVA Caffagni et al. (2024) train the model to generate answers with CLIP retrieval results. Different from Wiki-LLaVA, EchoSight (Yan & Xie, 2024) adopts a different training strategy and leverages the section-level retrieval annotation provided in E-VQA to train a reranking module.

**Performance Analysis.** To provide a comprehensive understanding of our proposed framework, we report its performance on the Encyclopedic VQA test set and the InfoSeek validation set. Our framework retrieves the essential knowledge from the text-only knowledge base WiKipedia (Vrandečić & Krötzsch, 2014) and utilizes the bunny-1.1 (He et al., 2024) and LLaMA3-8B (Dubey et al., 2024) to generate the answer. As shown in Table 1, the column labeled **LLM** indicates the employed large language model, while **KB** specifies what kind of knowledge base is used. The **Retrieval** describes the type of retrieved context categorized into two types: *KB Section* (the most relevant section of a Wikipedia page), and *KB Sentences* (specific paragraphs of knowledge content, representing the most challenging retrieval type). Finally, we report the BEM score on single-hop questions and all questions in the Encyclopedic VQA test dataset and the VQA score for unseen-question, unseen-entity, and all categories in the InfoSeek validation set.

From Table 1, we observe that the proposed framework significantly improves the performance of both large language models and multi-modal large language models without any additional training. For instance, the accuracy increase from *25.4* to *47.0* for Bunny-1.1 and from *10.6* to *47.5* for LlaMA3-8B in E-VQA. We also improve the VQA score in the InfoSeek such as *12.5* to *16.2* for Bunny-1.1 and from *1.2* to *15.2* for LlaMA3-8B.

Despite the strong performance, we have the following findings. First, our training-free methods outperform the fine-tuned baseline such as Wiki-LLaVA with plain CLIP retrieval results, indicating that the major challenge in knowledge-intensive VQA is the accuracy of knowledge retrieval. Another finding is that current state-of-the-art multimodal large language models or large language models are capable of understanding the retrieval results without the need for further fine-tuning on answer generation. Finally, the improvements of both our methods and EchoSight indicate the most urgent need is to improve knowledge retrieval in retrieval-augmented generation-based methods.

## 4.4 Ablation Study

In this section, we conduct an ablation study to provide a comprehensive understanding of each design in our proposed KIRA framework. Moreover, we provide the hyperparameters experiments.

**Impact of framework Design.** In the proposed framework, the multi-modal retrieval system for visual question answering is decomposed into multiple stages. To provide a deeper understanding of the effectiveness of each stage, we conducted ablation studies on the validation set of Encyclopedic VQA and randomly sampled 1,000 data from the InfoSeek validation set. As shown in Table 2, **CLIP I-T** indicates that naively utilizing CLIP to retrieve the knowledge base text with image, which serves as the baseline for our method. The **+FGM** means the fine-grained matching is applied on top of **CLIP I-T**, which improves the retrieval quality by involving the question relevance in the ranking of multiple knowledge articles. Finally, the **+DV** indicates that we add the text-retrieved

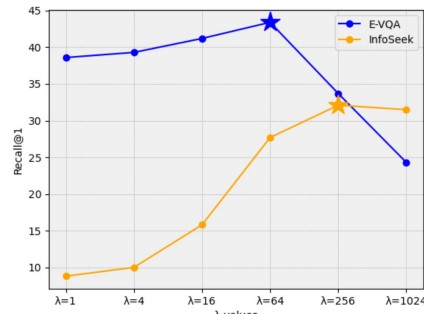

Figure 3: **The ablation study on hyperparameters.** We report the relationship between different values of and Recall@1 for E-VQA and the InfoSeek. The optimal value of is marked with a star. context in parallel with the visual-retrieved context from **+FGM**. Since two articles are retrieved, we only report the VQA performance under this setting.

From the results Table 2, we observe that the introduction of the *FGM* achieves significant improvements in E-VQA such as *106%* increase in recall@1, which demonstrates that including the question relevance besides the image relevance is important for correctly evaluating the relevance between visual question and the knowledge base. However, the improvements in the Infoseek are limited compared with the E-VQA. We consider the cause to be the relatively limited improvement space since the delta of recall@20 and recall@1 is *24.2* compared with *49.1* in E-VQA. For **+TRC**, we observe that the text-related context brings huge improvements to multi-hop questions in E-VQA, such as *33.2* to *41.1*, which indicates the question-only retrieval is important.

**Hyper Parameter Ablation.** The KIRA is a training-free multi-modal retrieval-augmented generation (RAG) system, requiring only lightweight hyperparameter tuning. We present the results of our ablation study concerning the weight $\lambda$. If the value of $\lambda$ increases, the importance of image relevance decreases. As shown in Table 3, we observe that the optimal choice for E-VQA is 64 and 256 for the InfoSeek, which indicates the sensitivity of the trade-off between the question relevance and image relevance changes across different datasets.

## 5 LIMITATION

Despite the promising results on knowledge-intensive VQA benchmarks without any training, our approach has several limitations. First, our method is a refinement for retrieval models, which means the upper bound of improvements is limited by the ability of the initial retrieval performance. Although we explore the possibilities of increasing the retrieval recall from recall@1 to recall@20 in this work, the proposed methods cannot be applied to improve recall@20. We consider this a more challenging task and will try to address it in the future. Another limitation is that the effectiveness of the proposed methods is potentially limited to certain scenarios since the improvement in E-VQA is significantly larger than that in the InfoSeek. We conclude that the mechanism of introducing detailed question-text relevance improves knowledge retrieval, whereas the top-ranking negative articles in the knowledge base do not contain essential information for the question.

## 6 CONCLUSION

In this paper, we introduced a novel retrieval-augmented generation (RAG) framework designed specifically to address the challenges of knowledge-intensive visual question answering (KI-VQA). Unlike traditional MLLMs that struggle with the fine-grained knowledge demands of KI-VQA tasks, our framework integrates general image-text similarity with detailed knowledge context to achieve precise entity recognition and effective knowledge retrieval. By employing a verification mechanism, we ensure that retrieved knowledge is relevant and sufficient to answer the posed questions, mitigating the limitations of global image-text matching approaches. Our experiments on the Encyclopedic VQA and the InfoSeek benchmarks demonstrate the efficacy of our approach, achieving significant improvements without the need for any fine-tuning or additional training. These results highlight the potential of our training-free, plug-and-play solution for KI-VQA tasks, offering a new pathway for integrating external knowledge bases into multimodal models. Moving forward, we believe that our framework opens up new possibilities for advancing the field of knowledge-intensive vision-language tasks, and we encourage future exploration in this direction.

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

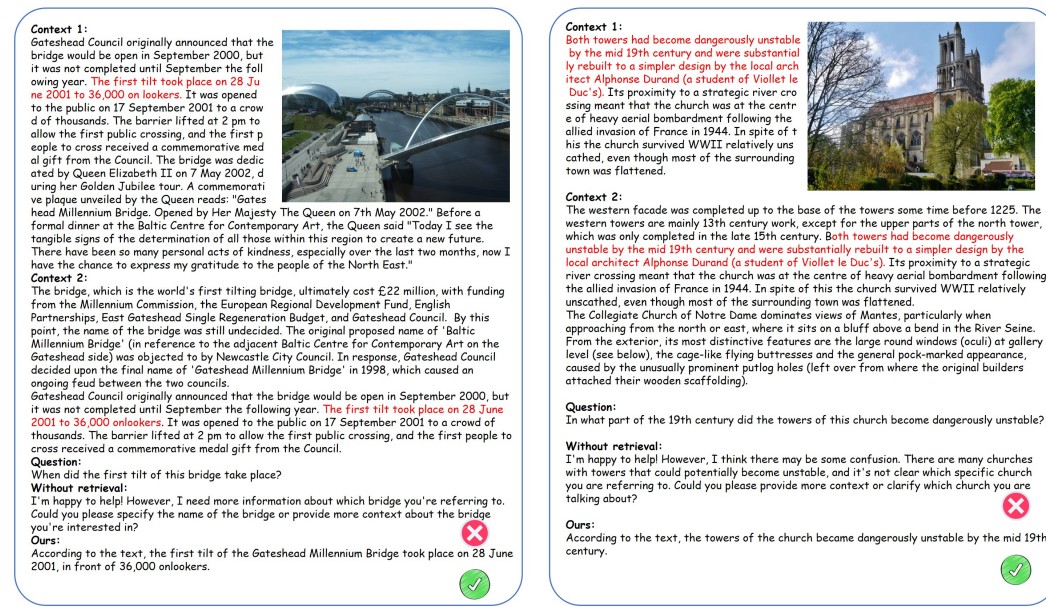

Figure 4: **The Visualization of the retrieval and answer generation.** We show the retrieval results of two samples in the Encyclopedic VQA dataset and provide the answer generated by the Bunny-1.1 with and without the retrieval results.

# A    APPENDIX

## A.1    QUANTITIVE VISUALIZATION

We provide qualitative results of our predictions. As shown in the Figure 4, the **Context1** is the $T\_visual_i^*$ and the **Context2** is the $T\_text_i^*$. The two retrieval results are extracted from distinct parts of the article in the WiKi knowledge base, which focus on the image and question. For the left image, the essential information is the time of the first tilt took place on the Gateshead Millennium Bridge. The retrieval system successfully found the WiKi article and we extracted two text chunks containing the "The first tilt took place on 28 Ju ne 2001 to 36,000 on lookers.". Without the retrieval results, the model fails to answer the question.

