# OpenReview forum: "Training-Free Retrieval-Augmented Generation for Knowledge-Intensive Visual Question Answering"
_ICLR.cc/2025/Conference — ICLR 2025 Conference Withdrawn Submission_

### Official Review · Reviewer_sGfr · 2024-11-01

**Soundness:** 2
**Presentation:** 2
**Contribution:** 2
**Rating:** 5
**Confidence:** 4

**Summary:**

The article introduces a novel, training-free retrieval-augmented generation (RAG) framework called "KIRA" that addresses challenges in knowledge-intensive visual question answering (KI-VQA). It highlights limitations in current multimodal large language models (MLLMs) for handling detailed knowledge tasks due to an overreliance on general image-text matching techniques. Instead, KIRA integrates general image-text similarity and fine-grained, question-specific relevance to improve knowledge retrieval and entity recognition without task-specific fine-tuning.

**Strengths:**

1)KIRA’s lack of need for fine-tuning increases flexibility, applicability across tasks, and minimizes resource demands.
2)By using both general image-text matching and a detailed question-context relevance check, KIRA enhances precision in entity recognition.
3)The results on Encyclopedic VQA and InfoSeek datasets underscore the framework's effectiveness without additional training requirements, improving practicality for real-world KI-VQA applications.

**Weaknesses:**

1)The paper's main contribution is proposing a straightforward, effective knowledge retrieval method. However, the experiments do not convincingly validate its effectiveness. In Table 1, comparisons are unfair, as training-free methods lack additional knowledge inputs, while other works, such as [1] demonstrates that simple retrieval methods can achieve strong results without training MLLMs. The paper should compare with more multimodal RAG approaches that also do not require training.
2)The paper emphasizes the "plug-and-play" nature of the method, but only tests it with LLAMA3-8B, which limits the scope of its effectiveness.
3)The ablation study is insufficient. For example, the paper employs both General Image-Text Matching and Detail Verification methods but does not include an ablation to test these individually.
4)Minor errors: The variable "N" in line 191 should be "M", and "FRA" in Table 1 might refer to "KIRA." Formatting issues include mistake quotation marks in LaTeX, the title "E-VQA" in Table 1 should replace "EVQA," and lines 444 and 445 are too close, affecting readability. Figure 1 is also somewhat inaccurate, as it omits image-text retrieval information in the fine-grained matching.
[1] Can Pre-trained Vision and Language Models Answer Visual Information-Seeking Questions?

**Questions:**

1)I’m curious about the reasoning behind certain retrieval metric settings. For example, in calculating T_visual, only the question is used to retrieve ​t_v. Why is the image not included at this stage? Given the parameter 𝜆 achieves good results when its value is large in coarse-grained retrieval, image information seems to be crucial. Additionally, the retrieval for T_text is overly coarse, as it searches the entire dataset. Does this approach not introduce noise information?
2)As in Weaknesses 1), how does this method compare with other training-free RAG methods? The paper only presents recall rates for CLIP I-T, but how would the RAG results vary with different CLIP retrieval configurations?
3)How does this method perform on other LLMs? Can it really be considered "plug and play" in other knowledge searches (such as searching outside of Wikipedia pages)?
4)What effect would a stronger retrieval model (e.g., a fine-tuned CLIP or language encoder other than ColBERT) have on the results?

---

### Official Review · Reviewer_Vhp9 · 2024-11-01

**Soundness:** 2
**Presentation:** 2
**Contribution:** 1
**Rating:** 3
**Confidence:** 4

**Summary:**

This paper introduces KIRA, a retrieval-augmented generation framework designed to improve multimodal large language models i.e. MLLMs on knowledge-intensive visual question-answering. Unlike task-specific fine-tuning approaches, KIRA enhances MLLMs through a training-free method, combining CLIP for general image-text similarity with a verification mechanism for question-text relevance. By doing so, KIRA aims to improve entity recognition accuracy in benchmarks such as Encyclopedic VQA and InfoSeek. The approach demonstrates performance gains without additional training. This work highlights the promise of a plug-and-play framework in addressing the challenges of knowledge-intensive VQA.

**Strengths:**

**Siginificant problem.** The paper addresses a significant problem of knowledge-intensive visual question-answering where the ability to handle specialized, nuanced information is essential.

**Training-free framework.** The proposed approach is training-free, which makes it adaptable and easily integrated with existing MLLMs. This plug-and-play nature not only would save time and resources but also offers flexibility in deployment, making it suitable for real-world settings.

**Weaknesses:**

**Limitations in the contributions.** The paper combines existing approaches such as RAG, CLIP, and LLMs, without any added value. Furthermore, for the fine-grained matching step, the framework relies on CLIP which aggregates data into a single token. This limits the ability of captioning detailed and localized information which is crucial for knowledge-intensive VQA. Since the framework is training-free and plug-and-play, I would recommend trying another more fine-grained model of your choice and analyzing its effectiveness.

**Lack of evaluation.** The evaluation scope is limited to only 2 VQA datasets. Other datasets, such as OK-VQA are also suitable for a visual question-answering setting where external knowledge is needed. The paper argues that KI-VQA “adds additional complexity to the task” compared to OK-VQA, so naturally, I would expect that a model suited for KI-VQA will perform well on OK-VQA. Also, it would be interesting to see how it will deal with standard VQA datasets, such as VQAv2, compared to existing baselines.

**Lack of visualizations and comprehensive presentation.** The paper lacks more visualizations, plotting attention maps, and qualitative analysis to improve the overall presentation. For instance, presenting visual results, such as outputs with accompanying retrievals and entity verification outcomes, could offer a more intuitive understanding of how the model processes the complex questions.

**Questions:**

- As mentioned in the weaknesses, could you evaluate your approach on the OK-VQA dataset, as well as standard VQA datasets, and compare it to existing baselines? This would help to explore the applicability of the KIRA framework to a larger scope of VQA tasks, including easier ones.

- How does the entity recognition module handle ambiguous cases where the entities may have similar appearances?

- What is the motivation for using ColBERT, instead of other text encoders? Is it possible to ablate this model choice?

- Minor comment: the space between lines 444 and 445 is too narrow.

- Minor comment: Line 458 mentions “Table 3" which is not present in the paper. I assume it should be “Figure 3” instead?

---

### Official Review · Reviewer_g6A1 · 2024-11-04

**Soundness:** 3
**Presentation:** 3
**Contribution:** 2
**Rating:** 3
**Confidence:** 4

**Summary:**

This paper addresses the challenge of relevant knowledge retrieval in Knowledge-intensive VQA tasks. The authors proposed to plug in a RAG framework on top of a CLIP-based Multimodal LLM to answer the visual questions. The proposed pipeline (KIRA) includes Wiki article retrieval followed with ranking among the candidate articles; then the answer is generated by a MLLM with the input of question and image and the two articles that are most relevant to the question visually and textually individually. Their proposed pipeline is evaluated on two KI-VQA benchmarks Encyclopedic VQA and InfoSeek, which demonstrates great improvements on both.

**Strengths:**

1. Overall, the key architecture and experiments are well presented in the paper. I found no trouble understanding the proposed pipeline, which should be attributed to the good organization of the paper. The technical details and implementation specifics are also clearly presented and easy to follow.

2. As stated by the authors, the proposed KIRA is training-free and the method may have the potential to be applied to other multimodal tasks.

**Weaknesses:**

1. The core mechanism is still built on similarity matching between inputs rather than true information comprehension. I would highly doubt the performance when the knowledge domain shifts or knowledge format changes, or on complex reasoning tasks that require deeper understanding.

Wiki articles varies a lot in terms of content and source across different entities, which inevitably introduces bias that cannot be addressed by the proposed pipeline. E.g. Q:  Where is this bird found? Given a picture of a Boissonneaua. The question can be fairly easy once the correct entity is retrieved. But a less commonly seen bird with few images or with an info page written in a completely different format is queried, then the model would likely to fail in the first step and produce cascading error.

2. The underlying limitation of training-free is that the performance would highly rely on pre-trained base models. The potential for information gain and performance improvement is limited by these models' pre-trained capabilities, which may vary significantly across different model choices. Though fine-tuning is avoided, hyper-parameters like $\lambda$ still need to be learnt for different datasets.

3. I believe the framework is built on top of some strong assumptions which may limit broader usage of the framework beyond the two benchmark datasets. For example, the framework assumes a very strong semantic alignment between questions and knowledge text, as enforced by initial matching and ColBERT for relevance verification, which means the performance are likely optimized for certain knowledge formats. I appreciate the authors efforts in evaluating the framework on the benchmarks, but the observations and insights might be less valuable to researches working with a different knowledge base or other forms of vqa.

**Questions:**

1. The representation leaning is quite imbalanced for visual and textual modality in the framework. Would the textual modality play a dominating role throughout the whole process? I would expect some more discussions on that topic.
2. How is the computational efficiency in retrieving Wiki articles? Any specific preprocessing or optimization done to improve retrieval efficiency? I'd like to see more details on the speed and computation overhead.
3. Have you evaluated answer quality with any other metrics besides accuracy?
4. The way of chunking knowledge may disturb the performance severely, as the training-free nature requires the representation of knowledge snippets to be aligned with CLIP output. Can you reveal more information on how you determine the size of text snippets, the distribution and how will that influence the retrieval phase?
5. How would the pipeline perform on general concepts for example have you benchmarked your proposed method on OKVQA?

---

### Official Review · Reviewer_j3NK · 2024-11-04

**Soundness:** 2
**Presentation:** 2
**Contribution:** 2
**Rating:** 3
**Confidence:** 4

**Summary:**

This paper introduces KIRA, a retrieval-augmented generation (RAG) framework designed to address knowledge-intensive visual question answering (KI-VQA) without task-specific fine-tuning of LLM. Utilizing CLIP for initial image-text matching and implementing a question-specific verification mechanism, KIRA combines general image-text relevance with question-text relevance to improve retrieval accuracy.

**Strengths:**

The proposed RAG approach does not require fine-tuning, presenting a training-free method that demonstrates notable performance improvements, suggesting its potential generalizability.

**Weaknesses:**

1. The integration of image relevance and question relevance is a common approach in knowledge-intensive VQA. The innovation here is incremental, mainly combining existing retrieval techniques.
2. As a training-free method, KIRA is tested only on LLaMA3-8B and a variant called Bunny-1.1, both based on LLaMA3. This limited scope raises questions about the broader applicability of KIRA to other model architectures.
3. Table 1 presents performance improvements, but comparisons are limited to vanilla models w/o RÅG and KIRA’s own results, without considering other retrieval-augmented generation methods. A more comprehensive baseline, including other RAG-based approaches, would provide a fairer assessment of KIRA’s performance.
4. The paper lacks references to recent advancements in RAG and RAG-based VQA, which could provide valuable context and positioning within the latest developments in the field.

**Questions:**

1. The current method combines two retrieval mechanisms without extensive exploration of potential design variations. Could there be a more integrated approach where image and question data are jointly leveraged for retrieval rather than processed separately?

---

### Official Review · Reviewer_oMcG · 2024-11-11

**Soundness:** 2
**Presentation:** 1
**Contribution:** 1
**Rating:** 3
**Confidence:** 4

**Summary:**

This paper introduces Knowledge-intensive Retrieval Augmentation (KIRA), a novel Retrieval Augmented Generation (RAG) framework designed for Knowledge-Intensive Visual Question Answering (KI-VQA). KIRA leverages existing Multimodal Large Language Models (MLLMs) to answer complex, knowledge-dependent visual questions without requiring task-specific fine-tuning. Its two-stage retrieval process first uses an image-text retrieval model to identify relevant knowledge, followed by a text retrieval model to refine the results based on fine-grained matching. KIRA comprises three core modules: Entity Recognition, Relevant Context Extraction, and Answer Generation. Experiments on Encyclopedic VQA and InfoSeek benchmarks demonstrate significant performance gains without additional training. Key innovations include KIRA's training-free, plug-and-play nature, its combined use of general image-text similarity and fine-grained context for precise entity recognition and knowledge retrieval, and its superior performance on established KI-VQA benchmarks.

**Strengths:**

In contrast to traditional RAG methods which typically rely on text for entity linking or topic identification, KIRA employs a two-stage multimodal entity recognition approach (coarse-grained CLIP retrieval followed by fine-grained ColBERT verification)，incorporating image information. Without additional training, it demonstrates performance improvements on knowledge-intensive VQA tasks.

**Weaknesses:**

1. The KIRA approach presented in this paper appears more like an engineering solution (CLIP and ColBERT), primarily involving the stacking of existing technologies, and thus demonstrating limited novelty. It also lacks in-depth theoretical analysis.
2. Many existing RAG systems can be applied to different knowledge bases and tasks with simple modifications or prompt engineering, also exhibiting a degree of "plug-and-play" capability. Therefore, the claim in the paper of being the "first plug-and-play" system is debatable.
3. The method offers limited parameter tuning space. KIRA primarily relies on the performance of existing models, possessing few adjustable parameters of its own, mainly focused on the λ parameter in Equation (6). This restricts its adaptability to different datasets and tasks.
4. The paper doesn't thoroughly discuss how the coverage and quality of the knowledge base affect retrieval results. Furthermore, the KIRA framework shows a much larger performance improvement on Encyclopedic VQA than on InfoSeek, suggesting that the method's applicability might be limited to certain scenarios.

**Questions:**

1. Is there a potential unfairness in the method comparison presented in Table 1? Given the variations in frameworks, models, and the type and coverage of the RAG knowledge bases used, shouldn't the performance of different RAG methods be compared solely on the same knowledge base for a more equitable evaluation?
2. Have you analyzed the computing resource requirements and efficiency issues of the KIRA framework in actual use?
3. Why is multimodality mentioned in the paper, but the knowledge base actually used only contains text, without considering other types of knowledge, such as images, videos, etc.

---

### Note · Authors · 2024-11-15

**Comment:**

Dear Editor and Reviewers,

We regret to inform you that we have decided to withdraw our submitted paper titled "Training-Free Retrieval-Augmented Generation for Knowledge-Intensive Visual Question Answering". We deeply appreciate the time and effort that the reviewers have invested in evaluating our work.

During the review process, we received valuable feedback that highlighted several areas for improvement. After careful consideration, we have concluded that addressing these points will require substantial revisions that are beyond the scope of the current submission. We believe it is in the best interest of the research community and our work to withdraw the paper at this stage.

We are grateful for the constructive comments and insights provided by the reviewers, which will undoubtedly help us in refining our research. We hope to resubmit a significantly improved version of our work in the future.

Thank you once again for your understanding and support.

**Withdrawal Confirmation:**

I have read and agree with the venue's withdrawal policy on behalf of myself and my co-authors.